# OpenReview forum: "LU-500: A Logo Benchmark for Concept Unlearning"
_ICLR.cc/2026/Conference — ICLR 2026 Conference Withdrawn Submission_

### Official Review · Reviewer_v8Vf · 2025-10-28

**Soundness:** 2
**Presentation:** 2
**Contribution:** 1
**Rating:** 2
**Confidence:** 5

**Summary:**

The paper proposes LU-500, a benchmark for “logo unlearning” in text-to-image models, built from 9,584 prompts across Fortune Global 500 brands with explicit (LUex-500) and implicit (LUim-500) tracks.

**Strengths:**

The benchmark design is clear. LU-500 isolates logo unlearning with two realistic prompting modes and a sizable, vetted prompt set.

**Weaknesses:**

1. I think this work essentially belong to the “prompt engineering” , not only LU-500 built from some prompts, but also the ProLU are three prompt-based LLM agents. Unfortunately, I do not see any algorithmic innovation in this work.

2. This work heavily relies on GPT-4o to built both the benchmark and the ProLU agents, yet Appendix A claims LLMs were used only to polish writing—that’s ridiculous

3. The author claims to “propose” 5 metrics as core contribution in the introduction, but CLIPScore and SSIM are common metrics and there is nothing new.

**Questions:**

See the weakness part of my review

---

### Official Review · Reviewer_xpRC · 2025-10-29

**Soundness:** 3
**Presentation:** 3
**Contribution:** 2
**Rating:** 4
**Confidence:** 4

**Summary:**

This paper propose a logo unlearning benchmark, LU-500 and also 5 metrics designed for quantitative evaluation of logo unlearning efficacy. Furthermore, a prompt-based unlearning method, ProLU, has been provided.

**Strengths:**

1. Well-motivated contribution. This paper fills a gap: prior benchmarks largely focus on natural images and stylistic concepts. Logos are intellectual property that diffusion model often memorize and highly relevant for both safety and legal compliance.
2. Proposed benchmark.The proposed LU-500 contains 500 logos across 10 commercial categories.
3. Four quantitative metrics are proposed for logo unlearning efficacy evaluation.

**Weaknesses:**

1. More strong concept unlearning baselines (e.g., [1] ) should be involved for benchmark evaluation.
2. Limited benchmark scale and diversity. Despite its value, 500 logos remain small relative to the variety of commercial marks. Many logos share similar geometric primitives, which might cause evaluation saturation.
3. Ambiguous boundary between memorization and semantic retention. Some metrics (e.g., CLIP-based LS) may not effectively differentiate between semantic similarity (“a bitten apple”) and literal logo reconstruction (“Apple Inc.” logo). A clearer delineation between concept-level leakage and pixel-level memorization would improve interpretability.

[1] Defensive Unlearning with Adversarial Training for Robust Concept Erasure in Diffusion Models, NeurIPS 2024

**Questions:**

Check the above weakness section.

---

### Official Review · Reviewer_UoF3 · 2025-11-01

**Soundness:** 2
**Presentation:** 2
**Contribution:** 2
**Rating:** 2
**Confidence:** 4

**Summary:**

This paper introduces a benchmark for logo unlearning. It focuses on evaluating inference time unlearning methods on this benchmark, and proposes a baseline unlearning method based on prompting editing. Through experiments, it shows that existing inference-time unlearning methods are not effective in unlearning logos.

**Strengths:**

The presentation of this paper is generally clear. And it provides an interesting correlation analysis between unlearning performance and various logo characteristics (area, location, edge density, etc.).

**Weaknesses:**

This paper has the following major limitations:
- The scope of copyright protection is limited. The exclusive focus on logos is too narrow for meaningful copyright protection. Other crucial copyrighted elements may include characters, protected artworks and patterns, and so on. The methods may not generalize to other types of copyrighted content
- The sole focus on inference-time unlearning methods is a major limitation because it does not represent the full spectrum of unlearning approaches. There is no explanation for why other unlearning approaches wouldn’t work. Fine-tuning methods and model manipulation unlearning should be compared with even if they might be more computationally demanding.
- The proposed baseline shows fundamental flaws. As we can see from Figure 6 that residual logos remain clear and brand identities are recognizable even if logos are partially removed. And it may not work if implicit brand indicators beyond logos are presented in the prompt.
- The current metric does not guarantee complete information removal. It does not test against adversarial attempts to recover logos.

The front page image needs some work, the layout is cluttered and does not clearly show the main information.

**Questions:**

Why focus exclusively on logos rather than a broader range of copyrighted visual content? Have you tested whether your benchmark and methods generalize to other types of copyrighted material (e.g., characters, artistic styles, patented designs)?

---

### Official Review · Reviewer_bZCH · 2025-11-04

**Soundness:** 3
**Presentation:** 3
**Contribution:** 2
**Rating:** 2
**Confidence:** 5

**Summary:**

The work introduces LU-500, a new benchmark designed to evaluate concept unlearning methods for company logos within text-to-image diffusion models.
The dataset contains prompts derived from Fortune Global 500 companies and is divided into explicit (LUex-500) and implicit (LUim-500) tracks.
The authors propose five quantitative metrics (CLIPScore, LogoScore, LogoSSIM, ImageScore, ImageSSIM) to assess both local logo removal and global image preservation across pixel and latent spaces.
Experiments compare inference-time unlearning methods (NP, SLD, SEGA) and fine-tuning approaches (ESD, Forget-Me-Not) on Stable Diffusion 3 Medium.
All baselines perform poorly, motivating the authors’ prompt-based baseline, ProLU, which edits prompts through a three-agent pipeline (Remover, Reflector, Checker). ProLU achieves stronger logo removal but somewhat weaker background preservation.
The paper also performs a correlation analysis between unlearning effectiveness and image characteristics (area, location, fractal dimension) and finds only weak relationships.

**Strengths:**

- New benchmark focusing on logo unlearning. This is a neglected but socially relevant copyright-protection task.
- The work is clearly written and easy to follow.
- The five proposed metrics systematically separate local logo removal from global fidelity, going beyond binary success rates.

**Weaknesses:**

- LU-500 focuses only on Fortune 500 logos; small-brand or non-Latin logos are not covered.
- Reliance on CLIP and SSIM metrics raises concerns about semantic leakage or bias: low CLIPScore may not perfectly reflect successful logo removal. Human evaluation or perceptual studies would strengthen claims of “logo removal.”
- The benchmark and metrics are valuable, but ProLU mainly repurposes LLM-based prompt rewriting without clear algorithmic innovation beyond dataset design.

**Questions:**

See weaknesses above.

---

### Note · Authors · 2026-03-01

I have read and agree with the venue's withdrawal policy on behalf of myself and my co-authors.

---

### Meta-Review · Area_Chair_gM7X · 2025-12-09

**Summary:**

Scope & generality
The scope of the benchmark is narrow: Fortune Global 500 logos only; little/no coverage of small brands, non-Latin scripts, or other copyrighted categories (characters, artworks, styles, patterns). (bZCH, UoF3, xpRC)

Novelty & contribution
Lack of algorithmic innovation: LU-500 is a dataset; ProLU is prompt-editing with LLM agents; CLIP/SSIM metrics are standard. (bZCH, v8Vf)
Heavy reliance on GPT-4o both to build the benchmark and to power ProLU; inconsistency with appendix claims about LLM usage. (v8Vf)

Evaluation design
Metrics: over-reliance on CLIP-based and SSIM-style scores; risk of semantic leakage or bias; need human/perceptual evaluation and adversarial tests for recovery. (bZCH, UoF3)

Baselines: stronger/unlearning baselines need to be added (e.g., recent concept-erasure methods), broader comparisons, and clearer separation of pixel-level memorization vs concept-level retention. (xpRC)


Method behavior
ProLU leaves residual, recognizable logos and struggles with implicit brand indicators. (UoF3)
LU-500 scale/diversity may cause saturation (many logos share primitives). (xpRC)

Presentation/clarity
Clarify exactly what is “proposed” vs reused (metrics, agents).
Clean up figures/front page; improve layout and documentation. (UoF3, v8Vf)

**Reviewer Concerns:**

As far as I could see, there is no rebuttal... So, there are only the original scores to go by (2,2,4,2) which suggest a reject decision.

**Reviewer Scores:**

N/A

---

### Decision · Program_Chairs · 2026-01-26

Reject